# Increase in the Bioactive Potential of Olive Pomace Oil after Ultrasound-Assisted Maceration

**DOI:** 10.3390/foods12112157

**Published:** 2023-05-26

**Authors:** Daniela Rigo Guerra, Lidia Betina Hendges Pletsch, Suelen Priscila Santos, Silvino Sasso Robalo, Stéphanie Reis Ribeiro, Tatiana Emanuelli, Daniel Assumpção Bertuol, Alexandre José Cichoski, Roger Wagner, Milene Teixeira Barcia, Cristiano Augusto Ballus

**Affiliations:** 1Department of Food Science and Technology, Federal University of Santa Maria (UFSM), Santa Maria 97105-900, Brazil; danielaguerraqmc@gmail.com (D.R.G.); pletschlidia@gmail.com (L.B.H.P.); suelenp_@hotmail.com (S.P.S.); silvinosr@gmail.com (S.S.R.); stephanieribeiro18@hotmail.com (S.R.R.); tatiana.emanuelli@ufsm.br (T.E.); cijoale@gmail.com (A.J.C.); rogerwag@gmail.com (R.W.); milene.barcia@ufsm.br (M.T.B.); 2Environmental Process Laboratory (LAPAM), Department of Chemical Engineering, Federal University of Santa Maria (UFSM), Santa Maria 97105-900, Brazil; dbertuol@gmail.com

**Keywords:** olive pomace oil, by-products, ultrasound, green technologies, response surface methodology, flavored oils

## Abstract

Olive pomace oil is obtained when a mixture of olive pomace and residual water is subjected to a second centrifugation. This oil has small amounts of phenolic and volatile compounds compared with extra-virgin olive oil. This study aimed to promote the aromatization of olive pomace oil with rosemary and basil using ultrasound-assisted maceration (UAM) to increase its bioactive potential. For each spice, the ultrasound operating conditions (amplitude, temperature, and extraction time) were optimized through central composite designs. Free fatty acids, peroxide value, volatile compounds, specific extinction coefficients, fatty acids, total phenolic compounds, antioxidant capacity, polar compounds, and oxidative stability were determined. After obtaining the optimal maceration conditions assisted by ultrasound, pomace oils flavored with rosemary and basil were compared to pure olive pomace oil. Quality parameters and fatty acids showed no significant difference after UAM. Rosemary aromatization by UAM resulted in a 19.2-fold increase in total phenolic compounds and a 6-fold increase in antioxidant capacity, in addition to providing the most significant increase in oxidative stability. Given this, aromatization by ultrasound-assisted maceration is an efficient method to increase, in a short time, the bioactive potential of olive pomace oil.

## 1. Introduction

In the processing of olives, the olive industry generates a large amount of waste and by-products, such as wastewater and olive pomace. This solid residue contains a high moisture content and fragments of pulp and fruit pits [1,2,3].

According to Vera et al. [4], the olives are crushed and cold-pressed in the modern method for extracting extra-virgin olive oil (biphasic form). According to the same authors, this process also gives rise to olive pomace, a solid residue that contains a large amount of residual oil.

Improper waste disposal has become an environmental concern, as these by-products have a phytotoxic effect and high biochemical oxygen demand [1,2,3]. Given this, the growing incentive to promote a “green” economy is combined with the need to develop strategies for treating waste from the olive industry and promote mechanisms to add value to these by-products [2,5].

In this context, the extraction of residual oil from olive pomace can be an exciting alternative in order to reduce the environmental impacts caused by its incorrect disposal and increase the income of olive producers who can benefit from the trade of an additional product.

Olive fruits have compounds of great nutritional interest in their composition, such as polyphenols, phytosterols, and dietary fiber. The residues resulting from the processing of olives are also characteristic of having biologically active constituents [3,5,6]. Much of the water-soluble fraction, including phenolic compounds, is transferred to the residue generated during the olive oil extraction [7]. According to Rodriguez-Rodriguez et al. [7], olive pomace oil can conserve fractions of triterpene compounds higher than those found in virgin olive oil. According to Rodriguez-Rodriguez et al. [8], hydrocarbons and sterols are also present in olive pomace oil and other compounds not found in olive oil.

Although olive pomace oil presents several compounds of interest, such as those mentioned above, it does not have organoleptic properties as attractive as those of extra-virgin and virgin olive oils. Therefore, it can be a product with the potential for innovation to improve its sensory and nutritional characteristics. Yilmazer et al. [9] state that olive oils flavored with spices have been increasingly highlighted since the aromatization process gives rise to differentiated products with alternative organoleptic properties that can attract more consumers. In addition to increasing the sensory quality of oils, aromatization promotes their enrichment with antioxidant compounds that migrate from herbs [10]. According to Kasimoglu et al. [11], some studies address consumer interest in olive oils infused with spices, including rosemary. According to the same authors, rosemary can be a natural antioxidant when added to olive oils. Furthermore, according to Sousa et al. [12], basil is another spice that can be used to promote the flavoring of olive oils and may also have antioxidant properties [13].

Usually, the aromatization of oils is carried out by maceration or contact aromatization. In this technique, the flavoring agent (herb, spice), which can be whole or ground, in natura or dry, is added directly to the oil, which may or may not be removed by filtration at the end of the process [14]. However, a trend in the flavoring area is the use of techniques that can accelerate the migration process of the compounds of interest in the flavoring agent to the oil. Among the technologies employed, “green” methods stand out. Among them, the most studied for oils so far include ultrasound and microwaves, with some works also evaluating pulsed electric fields and technologies with supercritical fluid [15].

Ultrasound, one of the “green” technologies, can encourage product aromatization. In addition to providing extractions in less time, it ensures excellent reproducibility and lower energy consumption than conventional extraction methods [15]. Shirsath et al. [16] report that the increased extraction capacity using ultrasound is related to the ultrasonic pressure waves propagated through the solvent and the generated cavitation effects (when bubbles are formed, they grow and subsequently break inside a liquid). The ultrasonic method has already been used in previous studies to promote the aromatization of olive oil with spices [17,18,19]. Our group conducted a study in which the ultrasound operating conditions were optimized for extra-virgin olive oil aromatization with rosemary and basil [20]. The results were auspicious, especially regarding the increase in the content of bioactive compounds and antioxidant capacity after aromatization, as well as the increase in oxidative stability. All this was obtained quickly due to the use of ultrasound.

Virgin olive oil has a bioactive and sensory quality even without adding spices. On the other hand, olive pomace oil has much smaller amounts of phenolic compounds and volatile compounds than olive oil. Thus, we hypothesize that the aromatization strategy of olive pomace oil with rosemary and basil, using ultrasound-assisted maceration, can significantly increase the bioactive and volatile compounds, causing the olive pomace oil to be converted into a product with more excellent added value and further stimulate its production from olive pomace. Quality parameters, total phenolic compounds, antioxidant capacity, polar compounds, oxidative stability, volatile compounds, and fatty acid contents were evaluated in the aromatized olive pomace oils to test whether their volatile and bioactive profiles improved. Scanning electron microscopy (SEM) was also used to study morphological changes in rosemary and basil particles after ultrasound-assisted maceration.

## 2. Materials and Methods

### 2.1. Samples and Reagents

Olive pomace oil (imported from Madrid, Spain) was purchased in Joinville (Santa Catarina, Brazil). The contents of the flasks were homogenized and stored in a plastic container at room temperature (approximately 25 °C) and away from light until the aromatization processes were carried out.

The homogenized olive pomace oil was analyzed for free fatty acid content (FFA), ultraviolet-specific extinction coefficients (K_232_, K_270_, and ΔK), peroxide index (PV), antioxidant capacity by ORAC method, total phenolic compounds (Folin–Ciocalteu reagent), oxidative stability (Rancimat method), polar compounds (TPC), antioxidant capacity (ORAC method), and fatty acid content (gas chromatography with flame ionization detector, GC- FID), and these procedures are described in detail in Section 2.3.

Rosemary (*Rosmarinus officinalis*) and basil (*Ocinum basilicum* L.) were purchased from a local trader in the city of Santa Maria (Rio Grande do Sul, Brazil). Before use, the moisture of the spices was determined. Samples were dried in a laboratory oven (105 °C) to determine moisture content until constant weight was attained. Both rosemary and basil had a moisture content of around 10%.

Chloroform p.a. (Neon, Suzano, SP, Brazil), glacial acetic acid p.a. (Neon, Suzano, SP, Brazil), sulfuric acid p.a. (Neon, Suzano, SP, Brazil), ethanol 95% p.a. (Neon, Suzano, SP, Brazil), soluble starch p.a. (Neon, Suzano, SP, Brazil), potassium biphthalate p.a. (Neon, Suzano, SP, Brazil), potassium iodate p.a. (Neon, Suzano, SP, Brazil), potassium iodide p.a. (Êxodo Científica, Sumaré, SP, Brazil), phenolphthalein p.a. (Neon, Suzano, SP, Brazil), monobasic potassium phosphate p.a. (Dinâmica, Indaiatuba, SP, Brazil), dibasic potassium phosphate anhydrous p.a. (Dinâmica, Indaiatuba, SP, Brazil), potassium hydroxide p.a. (Alphatec, São José dos Pinhais, PR, Brazil), methanol UV/HPLC (Dinâmica, Indaiatuba, SP, Brazil), sodium thiosulfate pentahydrate p.a. (Neon, Suzano, SP, Brazil), AAPH [2,2′-azobis(2-amidinopropane) dihydrochloride] (Sigma-Aldrich, St. Louis, MO, USA), sodium carbonate p.a. (Sigma-Aldrich, St. Louis, MO, USA), disodium fluorescein (Êxodo Científica, Sumaré, SP, Brazil), Folin–Ciocalteu reagent (Dinâmica, Indaiatuba, SP, Brazil), ethyl ether p.a. (Êxodo, Sumaré, SP, Brazil), gallic acid (Sigma-Aldrich, St. Louis, MO, USA), and Trolox (Sigma-Aldrich, St. Louis, MO, USA) were used in the experiments of this work.

### 2.2. Experimental Design for the Optimization of Ultrasound-Assisted Maceration

A central composite design (CCD) was chosen as a tool for the optimization procedure. It consisted of three variables in two levels (2^3^), with six axial points. The central point of the design was performed in quadruplicate. A separate CCD was performed for flavoring with rosemary or with basil. Each experimental design aimed to assist in optimizing the operating conditions of ultrasound and was composed of 18 experiments for rosemary and 18 for basil. Variables included ultrasound amplitude, which ranged from 50 to 100%, the temperature of the ultrasonic bath (from 15 to 35 °C), and extraction time (from 3.2 to 36.8 min), as seen in Table 1. Our research group has already successfully used this optimization protocol in previous work [20].

The ultrasonic equipment used was a 25 kHz fixed frequency ultrasonic bath (TI-H-10, Elma, Germany), operated in sweep mode.

The ultrasonic-assisted maceration (UAM) process was carried out in 250 mL Erlenmeyer flasks, adding 150 mL of olive pomace oil and 10% (m/m) of rosemary or basil. These ratios of rosemary and basil were previously evaluated in the study of Soares et al. [20]. Then, these flasks were taken to the ultrasonic bath, with the specific conditions of each experiment in the planning, in random order. After that, the contents of each Erlenmeyer flask were filtered under vacuum, through a paper filter, and then stored in an amber flask, protected from light and under refrigeration until subsequent analyses were carried out.

The responses chosen for the experimental designs were the quality parameters (free fatty acids, peroxide content, and specific extinction coefficients in the ultraviolet (K_232_, K_270_, and ΔK)); the profile of volatile compounds by gas chromatography with flame ionization detector (GC-FID) and gas chromatography coupled to mass spectrometry (GC-MS); oxidative stability using the Rancimat method; the content of polar compounds; the range of phenolic compounds by the Folin–Ciocalteu reagent method; and antioxidant capacity (ORAC method). The description of the procedures for these analyses can be found in Section 2.3.

For each response, mathematical models were calculated and validated using ANOVA (95% significance level) to verify whether the regression was significant and whether there was a significant lack of fit.

Furthermore, the optimal condition for each spice was calculated using all models simultaneously, using the Derringer–Suich algorithm. This algorithm is capable of combining all models to minimize or maximize results. As criteria for the simultaneous optimization, we set the reduction of free fatty acids, peroxide index, specific extinction coefficients (K_232_, K_270_, and ΔK), and polar compounds; and the maximization of the total range of phenolic compounds, antioxidant capacity, and oxidative stability of flavored oils. No criteria were used for volatile compounds.

After obtaining the optimal conditions for ultrasound-assisted maceration, these conditions predicted by the Derringer–Suich algorithm were experimentally executed in triplicate. Olive pomace oils flavored with rosemary and basil under optimal ultrasound conditions were subjected to determinations of free fatty acids, peroxide number, ultraviolet-specific extinction coefficients (K_232_, K_270_, and ΔK), oxidative stability, range of total phenolic compounds, the content of polar compounds, fatty acid content (by GC-FID), and antioxidant capacity.

Statistical analyses referring to experimental designs and prediction of optimal conditions were conducted using Design Expert 6.0.10 software (Minneapolis, MN, USA).

### 2.3. Procedures for Physicochemical Determinations

#### 2.3.1. Determination of Quality Parameters

Determinations of the quality parameters followed the official methods suggested by the European Union [21]. Each procedure is described briefly below.

Free fatty acid content: Approximately 2 g of the lipid sample was weighed in an Erlenmeyer flask. An amount of 50 mL of an ethyl ether/ethanol (1:1) solution was added and stirred. Two drops of phenolphthalein solution (1%, *w*/*v*) were added, and titration with 0.0095 mol·L^−1^ potassium hydroxide (previously standardized) was performed until the appearance of a pink color persisted for at least 10 s. The same procedure was performed for the blank. The free fatty acid content was calculated using the mass of oleic acid as a reference. The results obtained were expressed in the percentage of oleic acid (%).

Peroxide value: Approximately 2 g of the lipid sample was weighed in an Erlenmeyer flask with a ground screw. An amount of 10 mL of chloroform was added, and the sample was solubilized. An amount of 15 mL of acetic acid and 1 mL of saturated potassium iodide solution (12.8 g of potassium iodide in 10 mL of distilled water) were added. The Erlenmeyer flask was quickly shaken and kept in the dark for 1 min. Afterward, 50 mL of distilled water and 1 mL of aqueous starch solution (1%, *w*/*v*) were added. Titration was performed with 0.01 mol·L^−1^ sodium thiosulphate (previously standardized) until the dark-blue color disappeared. A blank test was carried out the same way as for the samples. The peroxide value was expressed in milliequivalents of active oxygen per kilogram of oil (meq O_2_·kg^−1^).

Ultraviolet-specific extinction coefficients: After carrying out different dilutions for the lipid samples with cyclohexane and proceeding with their complete homogenization, their absorbances were read at wavelengths of 232 nm (K_232_) and 270 nm (K_270_). The reading of absorbances at wavelengths of 266 nm and 274 nm was also performed and used to calculate the ∆K value. The dilutions were tested and carried out so that the absorbance reading did not exceed the detection limit of the spectrophotometer used.

Quality parameter analyses were performed in triplicate for pure olive pomace oil and olive pomace oils flavored with rosemary and basil and their optimal conditions.

#### 2.3.2. Fatty Acid Profile

Hartman and Lago performed fatty acid methylation using a transesterification method [22]. This method weighed 20 mg of lipid samples and added 1 mL of 0.4 mol·L^−1^ methanolic KOH solution. The mixture was vortexed and kept in a water bath at 100 °C for 10 min. After this step, 3 mL of 1 mol·L^−1^ methanolic solution of H_2_SO_4_ was added, and this mixture was vortexed and kept in a bath at 100 °C for 10 min. Subsequently, 2 mL of hexane was added, and vortexing was repeated. A centrifugation step was then performed.

Finally, the hexane phase containing fatty acid methyl esters (FAMEs) was injected into a 68900N Network GC System gas chromatograph (Agilent Technologies, Santa Clara, CA, USA) equipped with a flame ionization detector (Agilent Technologies, Santa Clara, CA, USA) and an automated injection system (G4513A). In this step, one μL of each sample was injected into a split–splitless injector, operating in split mode at a ratio of 1:50. The injector temperature was 250 °C. Analyses were performed on a DB-23 column (60 m × 0.25 mm × 0.25 µm) (Agilent, Santa Clara, CA, USA). Nitrogen was used as the carrier gas, with a constant pressure of 15.0 psi. The oven temperature program was as follows: initial temperature = 180 °C for 3 min; increasing the temperature to 200 °C, at a rate of 10 °C/min, maintaining this temperature for 20 min; temperature increase to 240 °C at a rate of 9 °C/min, maintaining this temperature for 15 min. The detector temperature was 280 °C. The detector gases were hydrogen (50 mL·min^−1^), nitrogen (24.6 mL·min^−1^), and synthetic air (300 mL·min^−1^). The FAMEs were identified by comparing the retention time of the sample compounds with the FAMEs standards (FAME MIX-37, Supelco, Bellefonte, PA, USA) and vaccenic acid. The results were presented as a percentage in the area of each FAME identified in the lipid portion about the total area of FAMEs in the chromatogram.

In triplicate, the fatty acid profile was determined for pure olive pomace oil and for the optimal points obtained after carrying out the experimental design for rosemary and basil.

#### 2.3.3. Profile of Volatile Compounds

Volatile compounds were extracted using headspace solid phase microextraction (HS-SPME), according to the technique proposed by Caporaso et al. [23], modifying the weight of the analyzed sample. An amount of 5 g of sample was weighed into a 20 mL PTFE flask closed with a silicone septum. The flasks containing the samples were placed in an immersion bath at 40 °C for 15 min for temperature equilibrium. Next, the fiber was exposed for 45 min using a DVB/Car/PDMS fiber (Divinylbenzene/Carboxen/polydimethylsiloxane) (2 cm, 50/30 μm, Supelco^®^, Bellefonte, PA, USA). Afterward, the samples were analyzed in a gas chromatograph equipped with a flame ionization detector (GC-FID; Varian 3400 Star, Palo Alto, CA, USA), and the identification was performed by a gas chromatograph coupled to a mass spectrometer (GC-MS; Shimadzu QP-2010 Plus, Tokyo, Japan). In the GC-FID, the fiber was introduced into the split–splitless injector operating in 1:30 split mode at 250 °C for 10 min. Compounds were separated on a ZB-WAX Plus column (30 m × 0.25 mm i.d.; 0.25 µm film thickness) (Phenomenex, Torrance, CA, USA). The GC-FID was operated with three gases: hydrogen, nitrogen (make-up), and synthetic air, whose flow rates were 30/30/300 mL·min^−1^, respectively. Hydrogen was used as carrier gas at a constant pressure of 103.42 kPa. A temperature program was carried out starting at 36 °C (maintained for 3 min), increasing to 80 °C at 5 °C/min (remaining for 2 min), later to 200 °C at 8 °C/min (for 2 min), and finally, a temperature of 230 °C was reached, increasing by 5 °C/min, remaining isothermal for 13 min. The detector was held at 250 °C.

The same injection conditions, column, and temperature program described for the GC-FID were employed to identify the volatile compounds by GC-MS. The detector was operated in EI (electron ionization) mode with an ionization energy of 70 eV, with a voltage of 0.99 kV, in scanning mode over a mass range of 35–350 *m*/*z*, with a cutoff of solvent at 1.5 min. As a carrier gas, helium was used at a constant flow rate (2 mL·min^−1^). The interface was maintained at 250 °C, and the ionization source at 230 °C. The compounds were identified with the aid of the NIST library, considering at least 85% of similarity, through the mass spectrum and the chromatographic experimental retention index of the compound. The experimental retention indices were calculated using a homologous series of alkanes (C_6_–C_24_) (Merck KGaA, Darmstadt, Germany) evaluated under these same chromatographic conditions.

#### 2.3.4. Assessment of Oxidative Stability

The oxidative stability of the lipid samples was determined using Professional Rancimat 892 equipment (Metrohm AG, Herisau, Switzerland), as described by the official method AOCS Cd 12b-92 (AOCS, 1997), with some modifications. Approximately 3 g of the samples were weighed in reaction flasks, then placed in the heating block at 150 °C with an airflow (filtered, clean, and dry) of 20.0 L·h^−1^. The volatile products originating from the degradation of the samples were collected in distilled water to measure their conductivity.

#### 2.3.5. Determination of Polar Compounds

The Testo 270 equipment (Testo, Germany) was used as a quality analyzer for cooking oils. This instrument provides the value of the total amount of polar materials present in lipid samples. In test tubes, a sufficient amount of sample was added so that the sensor of the equipment was wholly immersed in the lipid sample at the time of analysis (approximately 5 cm of sample in each test tube). The tubes were immersed in a water bath until 40 °C. Subsequently, the equipment was introduced into the sample tube, so the sensor was immersed in the oil. Then, after a few seconds, the TPM percentage was displayed on the equipment display.

#### 2.3.6. Determination of Total Phenolic Compounds and Antioxidant Capacity

##### Preparation of Polar Extracts for Analysis of Total Phenolic Compounds and Antioxidant Capacity

The polar extracts of the lipid samples were prepared following the study of Nakbi et al. [24]. A total of 2.5 g of sample was weighed into a centrifuge tube, and subsequently, 5.0 mL of hexane p.a. and 5.0 mL of methanol: water (60:40, *v*/*v*) were added. The mixture was vortexed for 2 min and then centrifuged for 5 min at 3000× *g*. The lower phase was separated and stored in flasks to be then subjected to determinations of total phenolic compounds and antioxidant capacity (ORAC method). All samples were extracted in triplicate (*n* = 3).

##### Total Phenolic Compounds

The total phenolic compounds content was determined by the Folin–Ciocalteu reagent method according to the procedure described by Singleton et al. [25]. In this procedure, 0.5 mL of polar extract of olive pomace oil was mixed with 2.5 mL of Folin–Ciocalteu reagent (diluted 1:10 in ultrapure water) and was then sheltered for 5 min. Then, 2 mL of 7.5% sodium carbonate was added, and the tubes were incubated in the dark for 2 h. Subsequently, the absorbance at 765 nm was measured. Quantitative results were calculated using an analytical gallic acid curve and were expressed as mg gallic acid equivalent per kg of oil sample (mg EAG kg^−1^). The analysis was performed in triplicate for the polar extracts. The calibration curve was constructed with 10 equidistant points with a gallic acid concentration ranging from 2 to 92 mg·L^−1^ (r^2^ = 0.9836).

##### Analysis of Antioxidant Capacity by the ORAC Method

The antioxidant capacity by the ORAC method (oxygen radical absorption capacity) was determined following the method presented by Ou et al. [26] and modified by Dávalos et al. [27]. The AAPH reagent (0.412 g) was completely dissolved in 10 mL of 75 mmol·L^−1^ phosphate buffer (pH 7.4), resulting in a final concentration of 152 mmol·L^−1^. The fluorescein working solution (81 nmol·L^−1^) was prepared minutes before the analysis by diluting the stock solution with 75 mmol·L^−1^ phosphate buffer (pH 7.4). The reaction was based on a mixture of 25 μL of polar extracts and 150 μL of fluorescein working solution in a 96-well polypropylene microplate. Then, the microplate was placed in the microplate reader and incubated for 10 min at 37 °C. Afterward, 25 μL of the AAPH radical was added. Fluorescence was recorded every minute for 120 min. The results obtained were used to calculate the area under the curve of the samples and the blank, making the difference between the two. The curve was constructed with eight equidistant points at different concentrations of Trolox (from 8 to 120 mg·L^−1^, r^2^ = 0.9783). The area under the curve was used to quantify the samples’ antioxidant capacity, and the results obtained were expressed as μmol equivalent of Trolox per g of sample (μmol TE g^−1^).

#### 2.3.7. Scanning Electron Microscopy (SEM)

The morphology of rosemary and basil samples was analyzed before and after the aromatization process using ultrasound. This analysis was performed using a scanning electron microscope (SEM) VEJGA-3G (Tecan, Czech Republic) with an accelerating voltage of 5 kV, a secondary electron (SE) detector for the rosemary samples, and a detector of secondary electrons (SE) plus a backscattered electron detector (BSE) for the basil samples. Before this analysis, the herbs submitted to the ultrasonic process were left in a desiccator on filter paper to remove excess oil. These samples were then previously metalized with gold in a current of 20 mA for 90 s.

### 2.4. Statistical Analysis

Results of the characterization analyses of pure pomace oil and the optimal conditions of ultrasound-assisted maceration were evaluated for normality (Shapiro–Wilk test) and homoscedasticity (Levene test) and then compared using the analysis of variance (ANOVA) and the Tukey test (*p* < 0.05). These statistical analyses were performed using the Statistica 7.0 software (StatSoft, Tulsa, OK, USA).

## 3. Results and Discussion

### 3.1. Optimization of Operating Conditions for Ultrasonic-Assisted Maceration (UAM) of Olive Pomace Oil

Appendix A present the results for the quality parameters (free fatty acids, peroxide content, and specific extinction coefficients), the range of total phenolic compounds, the antioxidant capacity, the range of polar compounds, the profile of volatile compounds, and the oxidative stability (Rancimat) for all the central composite design experiments.

Regarding the quality parameters, for the experimental design points referring to rosemary, a variation in the content of free fatty acids from 0.075% to 0.116% is observed (Appendix A). In comparison, for basil, this parameter ranged from 0.063% to 0.116% (Appendix A). The peroxide values for adding rosemary ranged from 2.96 to 6.97 mEqO_2_ kg^−1^ (Appendix A), while for basil, they ranged from 2.97 to 5.48 mEqO_2_ kg^−1^ (Appendix A). When the experimental design points are compared to pure olive pomace oil, the free fatty acid content was higher than the free fatty acid content of the 18 experiments referring to basil and rosemary. The peroxide content of flavored oils under different conditions was generally lower than the value observed for pure oil. For the experiments with the addition of basil (Appendix A), the K_232_ values ranged from 2.832 to 4.950, K_270_ values ranged from 1.275 to 3.486, and ΔK values ranged from 0.115 to 0.459. For the experiments with rosemary addition (Appendix A), K_232_ values ranged from 2.834 to 4.669, K_270_ values ranged from 1.220 to 1.463, and ΔK values ranged from 0.114 to 0.129.

Regarding the content of total phenolic compounds and antioxidant capacity (Appendix A), flavored samples showed higher values when compared to pure oil for all experiments, which suggests that phenolic compounds migrated from spices to oil.

The range for polar compounds (%TPM) was from 6.5 to 8.5 for the points referring to rosemary (Appendix A) and from 5.5 to 10.0 for the experimental points referring to basil (Appendix A). Regarding oxidative stability (Rancimat), these values were higher for all samples flavored with rosemary than for pure oil. For samples flavored with basil, there was a variation of 0.04 h to 1.18 h for the induction time and from 0.88 h to 1.55 h for the stability time.

The determination of the profile of volatile compounds identified 17 volatile compounds in the samples flavored with rosemary (Appendix A), and, for the samples flavored with basil, 19 compounds were identified (Appendix A). The complete list of volatiles identified in each experimental design and the range of observed contents can be seen in Figure 1.

It can be seen that the highest number of volatile compounds observed for the experimental design points whose samples were flavored with basil was experiment number 6 (17 compounds identified). At this point, the sample was exposed to an amplitude of 89.88% for 10 min at 30.95 °C. The smallest number of compounds identified was for experiment 1 (11 compounds observed). This experimental point had low time, temperature, and amplitude. It also can be seen that the lowest number of volatile compounds observed for the experimental design points whose samples were flavored with rosemary occurred for experimental points 2 (89.88%, 10 min, and 19.05 °C), 8 (89.88%, 30 min, and 30.95 °C), and 9 (50.0%, 20 min, and 25 °C), where the presence of 9 compounds was observed, and the highest number of volatile compounds (16) was observed for experiment number 4. Experiment 4 was exposed to an amplitude of 89.88% for 30 min at 19.05 °C.

The results obtained through the experimental design were used to model each response. After calculating the models, they were evaluated using ANOVA at a significance level of 95%. Linear, quadratic, or 2FI models were tested. Some models showed a significant lack of fit but with a p-value very close to the threshold. Thus, all the models were used to predict an ideal theoretical set of conditions for each spice. As the number of models and responses was high in each experimental design (27 models for rosemary and 29 models for basil), it was necessary to use the Derringer–Suich algorithm to combine all the models. Through this algorithm, it was possible to specify the maximization or minimization of some responses to obtain the optimal condition. It was defined that the algorithm maximized the content of total phenolic compounds, the antioxidant capacity, the induction time, and the stability time; and simultaneously minimized responses related to lipid oxidation (the content of free fatty acids, specific extinction coefficients, peroxide value, and polar compounds). The models for volatile compounds were the most affected by the lack of fit, and, therefore, it was decided not to define any criteria in the algorithm.

Thus, optimization using the Derringer–Suich algorithm gave us the ideal operating conditions for ultrasound-assisted maceration. For rosemary, the optimal condition consisted of 61.61% amplitude (−0.89), 28 min (0.80), and 16.97 °C (−1.35), with a desirability value of 0.60. For basil, optimal conditions comprised 50% amplitude (−1.68), 19.5 min (−0.05), and 18.39 °C (−1.11), with a desirability value of 0.75. The closer the desirability value is to 1.0, the closer the predicted condition is to the criteria used during simultaneous optimization.

The power generated by the ultrasonic bath was determined for each optimal condition. For rosemary, the power was 64 W, while for basil, the power was 37 W.

These optimal conditions were experimentally executed in triplicate, and their results were compared with those of pure pomace oil, as presented in the following section.

### 3.2. Comparison between Pure Olive Pomace Oil and Olive Pomace Oils Flavored under Optimal UAM

Figure 2 shows no significant difference between flavored and pure olive pomace oils for the quality parameters peroxide index, K_232_, K_270_, and ΔK. Only for the content of free fatty acids, the two flavored oils showed values slightly higher than those presented by the pure pomace oil. Thus, it can be stated that the aromatization process of olive pomace oil with rosemary and basil using the UAM process did not negatively affect the quality parameters. The parameter that evaluates the content of polar compounds (Figure 3) also showed no significant difference. Therefore, the procedure for calculating the optimal condition using the Derringer–Suich algorithm proved adequate, as the algorithm was asked to seek to minimize all parameters related to lipid oxidation.

Table 2 compares fatty acid contents between pure olive pomace oil and flavored oils. No significant difference was found in the content of the 11 fatty acids identified in the samples. This result also proves that using ultrasound in the UAM did not cause degradation of the olive pomace oil’s lipids.

As for the content of total phenolic compounds and antioxidant capacity, the difference was more significant (Figure 3). The rosemary UAM process in olive pomace oil increased the content of phenolic compounds by 19.2 times compared to crude pomace oil, while with the use of basil, the increase was 13.0 times. There was a 6.0-fold increase in the antioxidant capacity when using rosemary to flavor the pomace oil, while basil promoted a 2.0-fold increase in the antioxidant capacity.

The use of rosemary to carry out aromatization of olive pomace oil by UAM was the most efficient for increasing the content of phenolic compounds and antioxidant capacity, which was also reflected in the results for oxidative stability by Rancimat (Figure 3), where pomace oil flavored with rosemary showed the most extended induction and stability times.

These oxidative stability results are related to the higher content of total phenolics and antioxidant capacity in pomace oils with rosemary. It should be emphasized that when using the optimal condition of UAM, the aromatization of olive pomace oil with rosemary was carried out in just 28 min, which demonstrates that it is possible to increase the bioactive potential and oxidative stability of olive pomace oil, in addition to incorporating volatile compounds that will make it more pleasant for the consuming public, in a short time.

Olive pomace oil flavored with basil showed lower results than those observed for rosemary for total phenolics, antioxidant capacity, and oxidative stability, but still significantly higher than pure pomace oil. In this way, basil can also be considered an excellent alternative to increase olive pomace oil’s bioactive potential and phenolic compounds.

### 3.3. Scanning Electron Microscopy (SEM)

Through SEM, it was possible to visualize the physical effect of ultrasound on the structures of rosemary and basil before and after the ultrasonic procedure, as seen in Figure 4. The images were obtained from samples referring to the optimal conditions of each planning. This analysis was conducted to verify whether the ultrasonic method caused alterations on the surface of the analyzed spices, indicating matrix rupture and, consequently, the migration of compounds.

Through the highlighted regions, it can be observed that the exposure of herbs to the ultrasonic effect caused alteration in their morphological characteristics, with disruption of some structures, confirming that the spices were affected by the ultrasonic method, thus allowing the migration of compounds to olive pomace oil. This type of change in morphology has already been observed by Chemat et al. [15] and Khadhraoui et al. [28], as well as in our previous work with olive oil [20]. These authors say cavitation bubbles can be formed near the plant material’s surface during ultrasonic extraction. During compression cycles, the bubble ruptures, generating a microjet in the direction of the plant matrix. According to the same authors, factors such as the process’s high pressure and temperature can damage the plant’s cell walls, causing some compounds to be released into the medium.

Another interesting consequence of the differences between the structures of rosemary and basil was the difference in the power necessary to obtain the aromatization in optimal conditions. As stated before (Section 3.1), the power in the optimal condition for rosemary was almost double the power for the basil’s optimal condition. This can be correlated with the rosemary structure, which is more difficult to disrupt and needs more power to achieve morphological changes than is necessary for basil.

## 4. Conclusions

The aromatization process using ultrasound-assisted maceration, both with rosemary and with basil, promoted a significant increase in the content of total phenolic compounds, antioxidant capacity, and range of volatile compounds, as well as more excellent oxidative stability concerning crude pomace oil.

The ultrasonic method proved advantageous compared to conventional methods due to its speed (28 min for rosemary and 19.5 min for basil). The use of rosemary in the aromatization of olive pomace oil was responsible for the most significant increases in total phenolic contents, antioxidant capacity, and oxidative stability compared to basil.

Thus, it is concluded that promoting the aromatization of olive pomace oil can be an exciting strategy for olive oil producers since they may have the alternative of increasing the quality of the oil obtained from the by-product, helping to reduce the environmental impacts caused by the improper disposal of olive pomace. Given this, aromatization by ultrasound-assisted maceration is an efficient method when it is desired to increase, in a short time, the bioactive potential of olive pomace oil.

## Figures and Tables

**Figure 1 foods-12-02157-f001:**
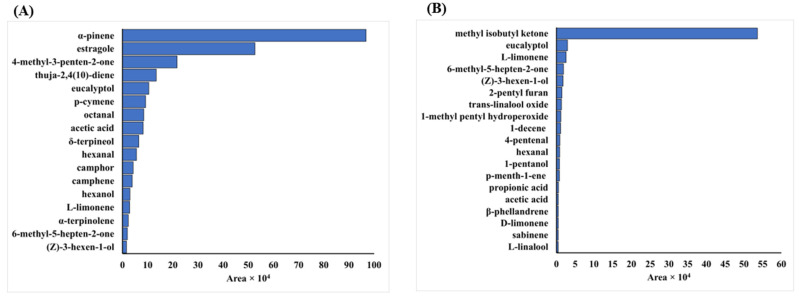
Range of volatile compounds observed in the experiments of the central composite design. (**A**) Volatile compounds identified in the experiments for rosemary; (**B**) Volatile compounds identified in the experiments for basil.

**Figure 2 foods-12-02157-f002:**
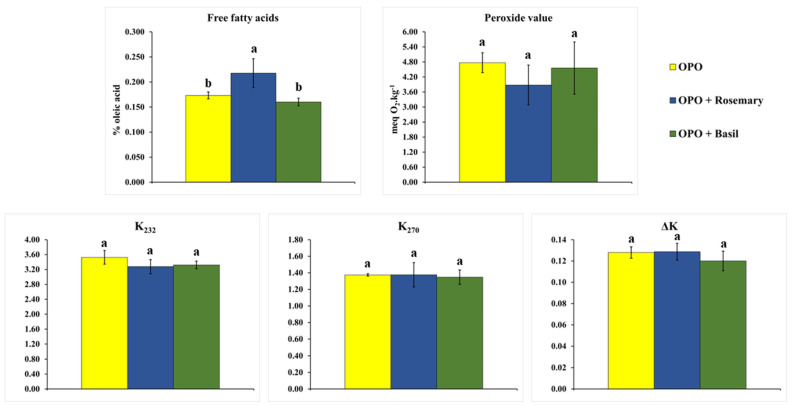
Quality parameters (free fatty acids, peroxide value, and specific extinction coefficients) comparison for olive pomace oil (OPO) and olive pomace oil aromatized with rosemary (OPO + Rosemary) and basil (OPO + Basil) after UAM. Means followed by the same letter showed no significant difference (*p* < 0.05) by the Tukey test.

**Figure 3 foods-12-02157-f003:**
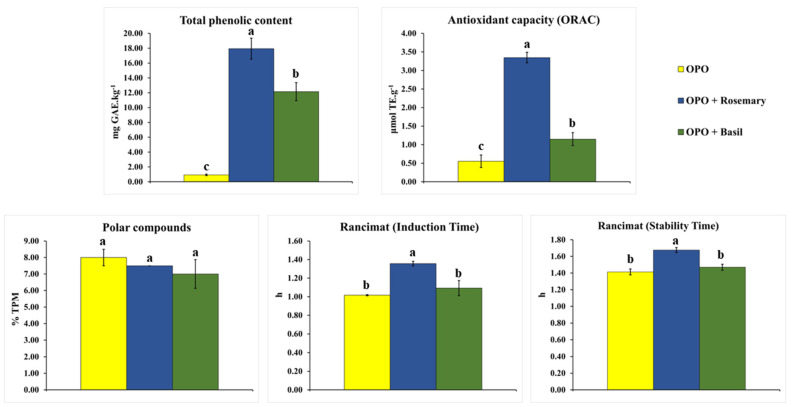
Total phenolic compounds, antioxidant capacity (ORAC), polar compounds, and oxidative stability (induction and stability times) comparison for olive pomace oil (OPO) and olive pomace oil aromatized with rosemary (OPO + Rosemary) and basil (OPO + Basil) after UAM. Means followed by the same letter showed no significant difference (*p* < 0.05) by the Tukey test.

**Figure 4 foods-12-02157-f004:**
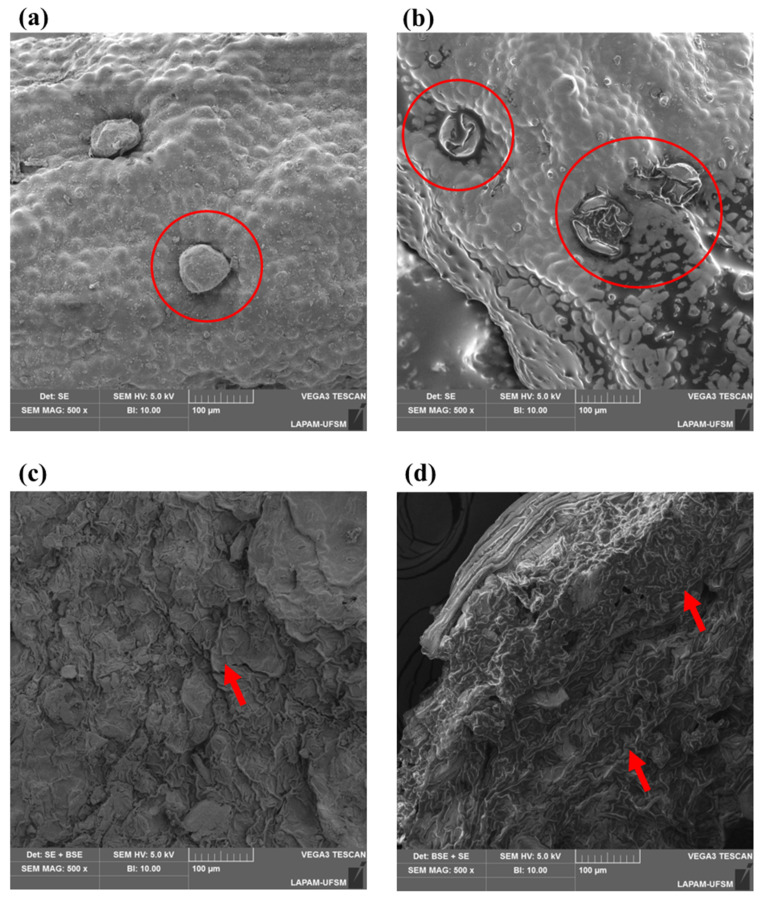
Scanning electron microscopy images: (**a**) rosemary before UAM (500×); (**b**) rosemary after UAM (500×); (**c**) basil before UAM (500×); and (**d**) basil after UAM (500×). Inside the ellipse in (**a**), an intact peltate glandular trichome in the rosemary surface; inside the ellipses in (**b**), there are disrupted peltate glandular trichomes in the rosemary surface after UAM. The red arrows in (**d**) denote a rugous and shattered surface in basil after UAM, when compared with the region pointed by the arrow in (**c**).

**Table 1 foods-12-02157-t001:** Matrix for the central composite designs (for rosemary and basil), considering coded and decoded variables.

Number of the Experiment	Coded Variables			Decoded Variables		
	x_1_	x_2_	x_3_	Amplitude (%)	Time (min)	Temperature (°C)
1	−1	−1	−1	60.12	10.0	19.05
2	1	−1	−1	89.88	10.0	19.05
3	−1	1	−1	60.12	30.0	19.05
4	1	1	−1	89.88	30.0	19.05
5	−1	−1	1	60.12	10.0	30.95
6	1	−1	1	89.88	10.0	30.95
7	−1	1	1	60.12	30.0	30.95
8	1	1	1	89.88	30.0	30.95
9	−1.68	0	0	50.00	20.0	25.00
10	1.68	0	0	100.00	20.0	25.00
11	0	−1.68	0	75.00	3.2	25.00
12	0	1.68	0	75.00	36.8	25.00
13	0	0	−1.68	75.00	20.0	15.00
14	0	0	1.68	75.00	20.0	35.00
15	0	0	0	75.00	20.0	25.00
16	0	0	0	75.00	20.0	25.00
17	0	0	0	75.00	20.0	25.00
18	0	0	0	75.00	20.0	25.00

**Table 2 foods-12-02157-t002:** Fatty acid composition for the olive pomace oil (OPO) and the optimal ultrasound-assisted maceration (UAM) conditions (mean ± standard deviation, *n* = 3).

Fatty Acids * (% Total Area)	Olive Pomace Oil (OPO)	Optimal UAM Condition for Rosemary in OPO	Optimal UAM Condition for Basil in OPO
16:0	11.592 ± 0.06 ^a^	11.602 ± 0.239 ^a^	11.510 ± 0.168 ^a^
16:1n-7	0.791 ± 0.01 ^a^	0.786 ± 0.027 ^a^	0.851 ± 0.069 ^a^
17:0	0.121 ± 0.03 ^a^	0.107 ± 0.020 ^a^	0.109 ± 0.023 ^a^
17:1n-7	0.139 ± 0.01 ^a^	0.159 ± 0.037 ^a^	0.166 ± 0.025 ^a^
18:0	2.867 ± 0.04 ^a^	2.834 ± 0.193 ^a^	2.799 ± 0.112 ^a^
18:1n-9	68.876 ± 0.55 ^a^	68.595 ± 1.080 ^a^	68.723 ± 0.594 ^a^
18:2n-6	11.100 ± 0.11 ^a^	11.027 ± 0.380 ^a^	11.338 ± 0.155 ^a^
18:3n-3	0.704 ± 0.02 ^a^	0.700 ± 0.059 ^a^	0.795 ± 0.066 ^a^
20:0	0.524 ± 0.02 ^a^	0.524 ± 0.045 ^a^	0.536 ± 0.055 ^a^
20:1n-9	0.348 ± 0.01 ^a^	0.339 ± 0.028 ^a^	0.321 ± 0.021 ^a^
22:0	0.264 ± 0.06 ^a^	0.244 ± 0.019 ^a^	0.285 ± 0.068 ^a^

* 16:0, palmitic acid; 16:1n-7, palmitoleic acid; 17:0, heptadecanoic acid; 17:1n-7, 10-heptadecenoic acid; 18:0, stearic acid; 18:1n-9, oleic acid; 18:2n-6, linoleic acid; 18:3n-3, α-linolenic acid; 20:0, arachidic acid; 20:1n-9, 11-eicosenoic acid; 22:0, behenic acid. Means followed by the same letter on the line showed no significant difference (*p* < 0.05) by the Tukey test.

## Data Availability

Data available on request.

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
