# Peer review of "Increase in the Bioactive Potential of Olive Pomace Oil after Ultrasound-Assisted Maceration"

_foods, 2023, doi:10.3390/foods12112157_

Round 1

Reviewer 1 Report

Utilization of olive pomace as olive pomace oil has high economic importance. The development of methods to increase its efficiency and economy has high practical relevance. This manuscript focuses on the applicability of ultrasound-assisted maceration (UAM) to increase the bioactivity in the aromatization of olive pomace oil with rosemary and basil.

Ultrasound is considered a promising technology to increase the efficiency of several processes for biomaterials, such as maceration, extraction, and drying/dehydration. The specific conditions of maceration using ultrasound are optimized by response surface methodology (RSM) using concentration of bioactive components and antioxidant properties as control parameters.

The manuscript is properly structured, with the introduction section summarizing the relevance of the study and the specific research motivations. The applied methods are suitable to test the hypothesis of the research and adequate to sample characteristics. The materials and methods are described clearly.

Although the manuscript contains interesting results, the discussion and analysis of the results are too superficial, in my opinion.

Comments and Suggestions:

        i.            The manuscript has strong descriptive feasibility. However, the results presented in section 3.1 and 3.2 are not discussed in details with relevant references.

      ii.            The energy efficiency and practical applicability of the UAM are not discussed in details compared to the 'conventional' process and other intensified methods. Please discuss the suitability for scale-up, etc.

Minor Comments:

[1.] I suggest reconsidering the keywords. Please include 'olive pomace oil' and 'ultrasound,' but delete 'GC-MS,' for instance.

[2.] Please briefly discuss other intensification methods (than UA) for enhanced maceration in the introduction section.

[3.] Please give the power of the sonicator (line 131)

[4.] Please explain how the rosemary/basil dosage was determined/selected (line 134).

[5.] In the title of section 2.3, 'in samples' is unnecessary.

[6.] The titles of sections 3.1 and 3.2 are too lengthy. Please remove 'the ultrasonic bath' and/or 'with rosemary and basil,' for instance.

[7.] Please rephrase the sentence in lines 318-322.

[8.] Please reconsider and shorten the title of Figure a. Also, give the titles as Figure 1.a and Figure 1.b in the figure caption to enhance the visibility and make the figures clear.

[9.] Please improve the visibility of axis titles in Figure 2 and Figure 3.

Reviewer 2 Report

The well-presented work is a valuable contribution to the valorization of olive oil by-products and thus to the contribution of research in terms of environmental sustainability in line with the goals of Agenda 2030.

In the introduction, the authors emphasize the importance and use of olive oil wastewater and olive pomace in the industry processes. There are considerable works in the literature that recall the health value of olive oil by-products, for example:

 Spizzirri, U.G.; Caputo, P. et al.  A Tara Gum/Olive Mill Wastewaters Phytochemicals Conjugate as a New Ingredient for the Formulation of an Antioxidant-Enriched Pudding. Foods 202211, 158. https://doi.org/10.3390/foods11020158

Therefore, it is strongly suggested to implement the cited work after references [1-3] also because it is much more recent than those reported.

In addition:

1.     I really appreciated the SEM study which is not usual in food chemistry work. Therefore, I suggest introducing SEM studies already in the section on the objectives of the paper given at the end of the introduction.

2.     It would be interesting to read in the text the table comparing the antioxidant profiles of pome olives with and without the addition of rosemary and basil.  I suggest including tables of the relevant results.

3.     It would be appropriate to include in the text the results from comparing the techniques used to enrich olive pomace with rosemary and basil. The results of the comparison between traditional maceration and ultrasonication would help to give a more usable result of the impact of the techniques at the environmental level.

4.     For the future, it would be interesting to initiate a study of the enriched extracts on cell lines so as to explore the biological activity of the extracts and possibly the synergy of the polyphenols they contain. In this regard, a cytotoxicity study is also suggested.

Reviewer 3 Report

Presently, olive oil production represents a valuable economic income for Mediterranean countries, where approximately 98 % of the world´s production is established. The cultivation of olive trees and the production of olive oil generate massive amounts of solid wastes, including olive tree pruning, olive leaves, olive stone, and liquid effluents such as olive pomace, olive oil mill wastewaters and olive mill waste, depending on the techniques used for oil manufacture. Olive by-products have long been considered as a challenging waste material impacting environmental protection, but this view has recently been reformed, leading to their recognition as a source of high-added value compounds as phenols and antioxidants that can be re-used as natural additives in the food, cosmetic and drug industries.

Ultrasound has become one of the most important techniques in green chemistry and emerging technologies. Many research investigations documented the usefulness of ultrasound in a wide range of application in food science, nanotechnology and complementary medicine, where effective extraction of natural products is important. Ultrasound treatment is an acoustic technology that can be used for non-invasive detection and/or modification of herbal bioactive compounds. As a physical treatment, it can modify the chemical and physical properties of biological systems at varying levels depending on the processing conditions (e.g., frequency, intensity and duration) and the herb´s structure and composition.

Herbs and spices are botanical raw materials that have active components that could be used in pharmaceutics, cosmetics, food additives, and health supplements. The herbs and spices have high contents of phenolic, carotenoid, flavonoid and volatile components that exhibit antimicrobial, antioxidant, and other biological activities.

This study aimed to promote the aromatization of olive oil pomace with rosemary and basil using ultrasound-assisted maceration to increase its bioactive potential. For each spice, the ultrasound operating conditions (amplitude, temperature and extraction time) were optimized through central composite designs. Quality parameters and fatty acids showed no significant difference after ultrasound-assisted maceration.

Recommendations

L 92  Please specify the storage temperature for olive oil pomace.

L 172 Please indicate the methods used to determine the quality parameters. Your phrase the official methods suggested by the European Commission (2016) is unclear, and there is no reference cited.

L 370  Please present the regression equation obtained via the Derringer-Suich algorithm. The correlation between responses and independent variables must be presented and explained.

Accept after minor revision

Round 2

Reviewer 1 Report

The manuscript has an interesting and relevant topic that can provide useful information for the practice, as well. The authors have revised the manuscript thoroughly according to reviewers' comments and suggestions. Additional information provided in the revised manuscript, rephrasing, and more detailed methodology and discussion part made the manuscript more complete and clear. The overall scientific quality of the manuscript has been improved significantly due to the revision. I agree and accept all modifications made by the authors.